# Erinacine A-Enriched *Hericium erinaceus* Mycelium Ethanol Extract Lessens Cellular Damage in Cell and *Drosophila* Models of Spinocerebellar Ataxia Type 3 by Improvement of Nrf2 Activation

**DOI:** 10.3390/antiox13121495

**Published:** 2024-12-07

**Authors:** Yu-Ling Wu, Hai-Lun Sun, Jui-Chih Chang, Wei-Yong Lin, Pei-Yin Chen, Chin-Chu Chen, Li-Ya Lee, Chien-Chun Li, Mingli Hsieh, Haw-Wen Chen, Ya-Chen Yang, Chin-San Liu, Kai-Li Liu

**Affiliations:** 1Cardiovascular and Mitochondrial Related Disease Research Center, Hualien Tzu Chi Hospital, Buddhist Tzu Chi Medical Foundation, Hualien 970, Taiwan; ylwu@tzuchi.com.tw; 2School of Medicine, Chung Shan Medical University, Taichung 402, Taiwan; cshy462@csh.org.tw; 3Department of Pediatrics, Division of Allergy, Asthma and Rheumatology, Chung Shan Medical University Hospital, Taichung 402, Taiwan; 4Center of Regenerative Medicine and Tissue Repair, Changhua Christian Hospital, Changhua 500, Taiwan; 145520@cch.org.tw; 5General Research Laboratory of Research Department, Changhua Christian Hospital, Changhua 500, Taiwan; 6Graduate Institute of Integrated Medicine, College of Chinese Medicine, China Medical University, Taichung 404, Taiwan; linwy@mail.cmu.edu.tw; 7Department of Medical Research, China Medical University Hospital, Taichung 40447, Taiwan; 8Department of Senior Citizen Welfare and Long-Term Care Business (Master Program), Hungkuang University, Taichung 433, Taiwan; peiyinchen@hk.edu.tw; 9Biotech Research Institute, Grape King Bio Ltd., Taoyuan 325, Taiwan; gkbioeng@grapeking.com.tw (C.-C.C.); ly.lee@grapeking.com.tw (L.-Y.L.); 10Department of Nutrition, Chung Shan Medical University, Taichung 402, Taiwan; licc@csmu.edu.tw; 11Department of Nutrition, Chung Shan Medical University Hospital, Taichung 402, Taiwan; 12Department of Life Science and Life Science Research Center, Tunghai University, Taichung 407, Taiwan; mhsieh@thu.edu.tw; 13Department of Nutrition, China Medical University, Taichung 404, Taiwan; chenhw@mail.cmu.edu.tw; 14Department of Health and Nutrition Biotechnology, Asia University, Taichung 413, Taiwan; yachenyang@asia.edu.tw; 15Vascular and Genomic Center, Institute of ATP, Changhua Christian Hospital, Changhua 50094, Taiwan; 16Department of Neurology, Changhua Christian Hospital, Changhua 500, Taiwan; 17Department of Post-Baccalaureate Medicine, College of Medicine, National Chung Hsing University, Taichung 402, Taiwan

**Keywords:** autophagy, erinacine A-enriched *Hericium erinaceus* mycelium, mutant polyQ-expanded ataxin-3, Nrf2, oxidative stress, PolyQ diseases, spinocerebellar ataxia type 3

## Abstract

Spinocerebellar ataxia type 3 (SCA3), caused by the abnormal expansion of polyglutamine (polyQ) in the ataxin-3 protein, is one of the inherited polyQ neurodegenerative diseases that share similar genetic and molecular features. Mutant polyQ-expanded ataxin-3 protein is prone to aggregation in affected neurons and is predominantly degraded by autophagy, which is beneficial for neurodegenerative disease treatment. Not only does mutant polyQ-expanded ataxin-3 increase susceptibility to oxidative cytotoxicity, but it also hampers antioxidant potency in neuronal cells. Nuclear factor erythroid-derived 2-like 2 (Nrf2), a master transcription factor that controls antioxidant and detoxification gene expression, plays a crucial role in neuroprotection in SCA3 and other neurodegenerative diseases. The present data showed that treatment with erinacine A-enriched *Hericium erinaceus* mycelium ethanol extract (HEME) extended longevity and improved locomotor activity in ELAV-SCA3tr-Q78 transgenic *Drosophila*. Moreover, HEME treatment enhanced antioxidant potency and autophagy, which, in turn, corrected levels of mutant polyQ-expanded ataxin-3 and restrained protein aggregation in both cell and *Drosophila* models of SCA3. Markedly, HEME increased the activation of Nrf2. Silencing Nrf2 protein expression negated most of the promising effects of HEME on SK-N-SH-MJD78 cells, highlighting the critical role of increased Nrf2 activation in the efficacy of HEME treatment. These findings suggest that HEME has therapeutic potential in SCA3 by enhancing autophagic and Nrf2-mediated antioxidant pathways, which may also influence neurodegenerative progression in other polyQ diseases.

## 1. Introduction

Polyglutamine (polyQ) diseases, including Spinocerebellar ataxias (SCAs) and Huntington’s disease (HD), are a group of late-onset, inherited genetic neurodegenerative disorders caused by abnormally long polyQ tracts, which are the result of expanded CAG repeats in different disease-related proteins [1]. The atypical expansion of the polyQ tract in these pathogenic proteins alters their functional characteristics, disrupting cellular homeostasis and causing various detrimental effects [1,2,3]. Spinocerebellar ataxia type 3 (SCA3), also known as Machado–Joseph disease, is the most prevalent polyQ SCA worldwide and is second only to HD in the incidence of polyQ diseases. SCA3 is triggered by the elongation of 61 to 87 glutamines in the polyQ tract near the C-terminus of the ataxin-3 protein, which typically has 10 to 44 glutamine repeats in unaffected individuals [4]. Similar to the pathogenic proteins of other polyQ diseases, the mutant polyQ-expanded ataxin-3 protein selectively triggers neuronal cell death in the cerebellum, striatum, and pons. This results in progressive cerebellar ataxia, ophthalmoplegia, dysarthria, dysphagia, dystonia, rigidity, distal muscle atrophies, and the premature death of affected SCA3 patients [4]. To date, no effective treatments have been approved to stop or delay the progression of SCA3 or other polyQ diseases.

Although not the only factor, the mutant polyQ-expanded ataxin-3 protein, which is susceptible to misfolding and aggregation, is substantially associated with neuronal cell death in SCA3 patients [4,5]. Autophagy, a lysosome-dependent degradation pathway, is regarded as a key mechanism for the intracellular clearance of misfolded protein aggregates. Enhancing this process may offer therapeutic benefits for neurodegenerative diseases, including polyQ diseases [3,6]. The expression of the autophagy initiation protein beclin-1 was found to be decreased in postmortem brains of SCA3 patients [7]. Induction of autophagy and overexpression of beclin-1 reduced mutant polyQ-expanded ataxin-3 expression and aggregation in the cerebellum and also ameliorated Purkinje cell loss, motor impairment, and the ataxic phenotype in cell and transgenic animal models of SCA3 [8,9,10,11]. These findings underscore the importance of autophagy as a therapeutic target in SCA3.

The progression of oxidative stress, caused by reactive oxygen species (ROS) and free radicals, is considered to contribute to the initiation and development of numerous late-onset diseases, including cancer and neurodegenerative disorders [12]. Diminished endogenous antioxidant defenses lead to increased oxidative stress, which, in turn, produces neural cellular toxicity in SCA3 and other polyQ diseases [1]. Furthermore, mutant polyQ-expanded ataxin-3 and huntingtin result in decreased transcriptional activity of nuclear factor erythroid 2-related factor 2 (Nrf2), the principal transcription factor that binds to and transcribes a series of antioxidant and cytoprotective gene expression with antioxidant response element (ARE) sequences in their promoter regions [13,14]. In addition to regulating oxidative stress, the Nrf2 pathway is also implicated in the reduction in mutant polyQ-expanded ataxin-3 protein expression and aggregation in SCA3 cell models [14,15,16]. Consequently, activating the Nrf2 pathway may be a potential therapeutic target for delaying or arresting SCA3 and other polyQ neurodegenerative diseases.

A large number of experimental and epidemiological studies support the use of food-derived products for disease prevention and complementary treatments due to their safety and fewer side effects compared to synthetic drugs [17]. *Hericium erinaceus* (*H. erinaceus*), commonly known as lion’s mane mushroom, Yamabushitake, or monkey’s head mushroom, belongs to the family *Hericiaceae* and is used as a medicinal–culinary mushroom in regions such as Taiwan, Korea, Japan, and China [18,19,20]. Many active compounds isolated from the fruiting body or mycelium of *H. erinaceus* display diverse medicinal properties, including hemagglutinating, immunomodulatory, hypolipidemic, antihyperglycemic, antimicrobial, antitumor, and antioxidant effects [20]. Furthermore, studies have shown that *H. erinaceus* mycelia, enriched with erinacine A, the major bioactive cyathane diterpenoid in the mycelia, exhibit significant anti-neurodegenerative and neuroprotective activities [18,19]. Recently, our study demonstrated that erinacine A-enriched *H. erinaceus* mycelium ethanol extract (HEME) protected against pro-oxidant tert-butyl hydroperoxide-induced neuronal cell apoptosis in experimental models of SCA3 [21]. To further explore the therapeutic potential of HEME in SCA3, we used human neuroblastoma SK-N-SH-MJD78 cells and ELAV-SCA3tr-Q78 *Drosophila* expressing full-length ataxin-3 with 78 CAG repeats and an ataxin-3 polyQ tract of 78 residues, respectively, to assess the beneficial impacts and possible molecular mechanisms of HEME against mutant ataxin-3-induced neuronal damage.

## 2. Materials and Methods

### 2.1. Materials

The human neuroblastoma cell line SK-N-SHMJD78, stably expressing full-length ataxin-3 with 78 CAG repeats, was obtained from Prof. Mingli Hsieh (Department of Life Science, Tunghai University, Taichung, Taiwan). The UAS-MJDtr-Q27, UAS-MJDtr-Q78, and *elav*-Gal4 fly strains were acquired from the Bloomington *Drosophila* stock center (Indiana University, IN, USA). The HEME was gifted by Biotech Research Institute, Grape King Bio Ltd. (Taoyuan, Taiwan). The erinacine A content in the HEME was 5 mg/g, and the incubation, extraction, and high-performance liquid chromatography analysis of HEME were described earlier [21]. Antibodies against heat shock protein 27 (Hsp27, sc13132), NAD(P)H: quinone oxidoreductase 1(NQO-1, sc16464), Nrf2 (sc722), Beclin-1 (sc11427), and glyceraldehyde 3-phosphate dehydrogenase (GAPDH, sc32233) were from Santa Cruz Biotechnology (Santa Cruz, CA, USA). Specific antibodies for manganese superoxide dismutase (SOD1, GTX100659), SOD2 (GTX116093), catalase (GTX110704), glutathione peroxidase (GPx1, GTX116040), GPx2 (GTX100292), and glutathione reductase (GR, GTX114199) were obtained from GeneTex Inc. (Alton Pkwy Irvine, CA, USA). Antibodies for poly ADP-ribose polymerase (PARP, #9542) were purchased from Cell Signaling Technology Inc. (Beverly, MA, USA). Specific antibodies for microtubule-associated protein 1 light chain 3 (LC3, M186-3), Heme Oxygenase-1 (HO-1, 374090), and β-actin (#MAB1501) were from MBL International (Woburn, MA, USA), Calbiochem (San Diego, CA, USA), and Millipore (Billerica, MA, USA), respectively. The specific antibodies to p62 (ab56416) and ataxin-3 (ab175265) were from Abcam (Cambridge, MA, USA). The PROTEOSTAT^®^ Protein aggregation assay was obtained from Enzo Life Science (Farmingdale, NY, USA).

### 2.2. Drosophila Stocks and Crosses

UAS-MJDtr-Q27 and UAS-MJDtr-Q78 fly stocks selectively express ataxin-3tr-Q27 or ataxin-3tr-Q78 in the nervous system and were bred as described in our previous study [21].

### 2.3. Longevity and Locomotor Activity in ELAV-SCA3tr-Q27 and ELAV-SCA3tr-Q78 Transgenic Drosophila

After eclosion, female flies were grown on the standard media added with or without DMSO vehicle control, 0.5 or 1% HEME. The medium was changed and the survival rate and locomotor activity were assessed every 3 days. The longevity was graphed and compared by using Kaplan–Meier log rank survival statistics in SigmaStat V3.5 software (Systat Software, Inc., San Jose, CA, USA). For evaluation of locomotor activity, flies were tapped down and the number of flies reaching 5 cm in 18 s was recorded. The locomotor activity (%) was determined as *N*_top_/*N*_total_ × 100, where *N*_total_ and *N*_top_ represent the number of total flies and the number of flies at the top (over the 5 cm line), respectively.

### 2.4. Cell Culture and Treatment

The neuroblastoma SK-N-SH-MJD78 cells (5 × 10^5^ cells/mL) were maintained as described in our previous study [21]. When the cells reached 90% confluence, they were treated with dimethyl sulfoxide (DMSO) as the vehicle control or with 1.25–5 µg/mL HEME and were analyzed as designated in the figure legends.

### 2.5. Cell Viability Assay

After treatment for 24 h, a 3-(4,5-dimeth-ylthiazol-2-yl)-2,5-diphenyltetrazoliumbromide (MTT) assay was employed as an indicator of cell viability as described in our previous study [21]. The absorbance of the supernatant from each sample was read at 570 nm in a VersaMax Tunable Microplate Reader (Molecular Devices Corporation, Sunnyvale, CA, USA).

### 2.6. Protein Extraction and Western Blot

Radioimmunoprecipitation assay buffer as well as hypotonic and then hypertonic extraction buffer were used to isolate total protein and nuclear protein extracts, respectively [22]. Equal amounts of total and nuclear proteins (15–30 μg) were used for the Western blot experiment [21]. Proteins were imaged using specific antibodies and horseradish peroxidase-conjugated anti-mouse (sc516102) or anti-rabbit IgG (sc2357) (Santa Cruz). Notably, the molecular weight of normal ataxin-3 is 42 kDa, and due to protein aggregation, mutant ataxin-3 displays a distinct band at 73 kDa. Immunoblots were visualized and quantified as described in our previous study [21].

### 2.7. Measurements of Mitochondrial ROS, Autophagic Cells, and Protein Aggregation

Levels of ROS in cell and fly models of SCA3 were determined with CM-H2DCFDA and Invitrogen™ MitoSOX™ (Thermo Fisher Scientific, Waltham, MA, USA). Acridine orange staining and ProteoStat^®^ protein aggregation assay were used to quantify autophagic cells and protein aggregation (Thermo Fisher Scientific). Fluorescence intensity was measured by use of a FlexStation^®^3 Multi-Detection Reader (Applied Biosystems, Foster City, CA, USA).

### 2.8. Measurements of Glutathione (GSH) Content

After treatment for 24 h, cells were collected on iced PBS. GSH content in SK-N-SH-MJD78 cells was measured using a GSH+GSSG/GSH assay kit (Abcam) as per the manufacturer’s instructions. Colorimetric intensity at 405 nm was measured using a CLARIOstar Plate Reader (BMG LabTech, Ortenberg, Germany).

### 2.9. Measurements of Catalase, GPx, and SOD Activity

After treatment for 24 h, cells were collected on iced PBS. Catalase, GPx, and SOD activity were assayed in SK-N-SH-MJD78 cells with Amplite^®^ Fluorimetric Catalase, Glutathione Peroxidase, Superoxide Dismutase assay kits (AAT Bioquest, Pleasanton, CA, USA), as per the manufacturer’s instructions. Fluorescence intensity was measured using a CLARIOstar Plate Reader (BMG LabTech).

### 2.10. Plasmids and Transient Transfection

The pGL3 promoter–luciferase plasmid expressing a 2xARE fragment containing tandem repeats of double-stranded oligonucleotides spanning the Nrf2 binding site, 5′-TGACTCAGCA-3′, was used to measure Nrf2 transcriptional activity [23]. Cells were transiently transfected with plasmid (0.1 μg/mL) or control vector (0.3 μg/mL) by use of Lipofectamine 2000 transfection reagent (Thermo Fisher Scientific).

### 2.11. Reporter Gene Assay

Nrf2 transcriptional activity was determined and corrected based on β-galactosidase activity using a Luciferase Assay System and a β-Galactosidase Enzyme Assay System from Promega Co. (Madison, WI, USA).

### 2.12. Nrf2 Small Interfering RNA (siRNA) Transfection

To knock down Nrf2 expression, MJD78 cells were transfected with an siRNA targeting Nrf2 and non-targeting control (NTC) siRNA (6 nM, Thermo Fisher Scientific) using Lipofectamine RNAiMAX transfection reagent. After 16 h of transfection, the cells were treated with HEME as described in the figure legends.

### 2.13. Statistical Analysis

Data are expressed as the means ± SDs from at least three independent experiments. Student’s *t*-test was used to examine the statistical differences between MJDtr-Q78 and ELAV-MJDtr-Q27 transgenic flies treated with vehicle control. In MJDtr-Q78D transgenic flies and SK-N-SH-MJD78 cells, differences among treatments were examined statistically using a Statistical Analysis System (Cary, NC, USA) by one-way ANOVA and Tukey’s multiple-range test.

## 3. Results

### 3.1. Effects of HEME on Longevity and Locomotor Activity in ELAV-SCA3tr-Q78 Transgenic Drosophila

In agreement with previous data, our results showed that the lifespan and climbing activity of MJDtr-Q78 transgenic flies were diminished compared to ELAV-MJDtr-Q27 control flies [14,21]. These declines were prevented by the administration of 0.5% and 1% HEME, compared to the ethanol vehicle control (*p* < 0.001, Figure 1 and Figure 2).

### 3.2. Effects of HEME on Protein Aggregations and Expression of Mutant Ataxin-3 and Hsp27 in ELAV-SCA3tr-Q78 Transgenic Drosophila

The pathogenic mutant polyQ-expanded ataxin-3 protein tends to accumulate and aggregate, leading to cerebellar dysfunction [4,5]. Compared to the parental SK-N-SH cells, Hsp27 expression was decreased in SK-N-SH-MJD78 cells. The overexpression of Hsp27 mitigated susceptibility to polyQ toxicity in SK-N-SH-MJD78 cells [24]. MJDtr-Q78 transgenic flies exhibited increased levels of mutant polyQ-expanded ataxin-3 and protein aggregation, along with decreased Hsp27 expression, compared to ELAV-MJDtr-Q27 control flies. The administration of HEME could address these observed deficiencies (Figure 3A, *p* < 0.001 and Table 1, *p* < 0.0001).

### 3.3. Effects of HEME on Autophagy and Nrf2-Mediated Signaling in ELAV-SCA3tr-Q78 Transgenic Drosophila

As verified in our previous study [14], ELAV-SCA3tr-Q78 flies exhibited increased oxidative stress and decreased autophagy compared to ELAV-MJDtr-Q27 control flies. Notably, there was a reduction in oxidative events, including declines in total and mitochondrial ROS levels, along with increased expression of Nrf2 and antioxidant enzymes such as HO-1, NQO-1, GR, catalase, GPx1, GPx2, and SOD1 and SOD2 proteins in ELAV-SCA3tr-Q78 flies fed with HEME (*p* < 0.001, Figure 3B and *p* < 0.0001, Table 1). Moreover, the addition of HEME enlarged the expression of autophagy-related proteins such as p62, beclin-1, and LC3-II in ELAV-SCA3tr-Q78 flies (*p* < 0.001, Figure 3A).

### 3.4. Effects of HEME on Mutant Ataxin-3, Hsp27, and Protein Aggregation in SK-N-SH-MJD78 Cells

HEME treatments have no adverse effects on the growth of SK-N-SH-MJD78 cells (Table 2). Consistent with the in vivo data, HEME treatment, compared to the DMSO vehicle control, diminished mutant polyQ-expanded ataxin-3 protein expression and protein aggregation levels and enhanced Hsp27 expression without affecting cell viability in SK-N-SH-MJD78 cells (*p* < 0.0001, Figure 4 and Table 2).

### 3.5. Effects of HEME on Autophagy in SK-N-SH-MJD 78 Cells

Autophagy, a lysosomal degradation pathway that facilitates the clearance of mutant polyQ-expanded ataxin-3 and its aggregation [8,9,10,11], is declined in SK-N-SH-MJD78 cells compared to SK-N-SH-WT and SK-N-SH-MJD26 cells [14]. Compared to the DMSO vehicle control, treatment with HEME in SK-N-SH-MJD78 cells augmented autophagy levels, as evidenced by increases in protein expression of p62, beclin-1, and LC3-II, as well as the amount of the lysosomotropic agent acridine orange (*p* ≤ 0.0001, Figure 5).

### 3.6. Effects of HEME on Oxidative Stress, GSH Levels, and Expression and Activity of Nrf2 and Antioxidant Protein in SK-N-SH-MJD 78 Cells

In vivo and in vitro studies suggest that oxidative stress contributes to polyQ protein aggregation and neuronal cell death in SCA3 and other polyQ diseases [1]. HEME treatment protected against oxidative stress in SK-N-SH-MJD78 cells, as determined by reductions in H2-DCFDA levels and increases in GSH levels, as well as enhanced expression of antioxidant enzymes and Nrf2 nuclear proteins (*p* < 0.001 and H2DCFDA; *p* = 0.012, Figure 6; *p* < 0.0001, Figure 7). Additionally, GPx, catalase, and SODs enzyme activities, along with Nrf2 transcriptional activity, were amplified in SK-N-SH-MJD78 cells treated with HEME (*p* < 0.001, Figure 6 and *p* < 0.0001, Figure 7).

### 3.7. Improving Nrf2 Activation Is Important in the Beneficial Effects of HEME on SK-N-SH-MJD78 Cells

Nrf2 siRNA was used to examine the role of Nrf2 activation in the health benefits of HEME in SK-N-SH-MJD78 cells. Western blot analysis and the reporter gene assay revealed that Nrf2 siRNA effectively knocked down Nrf2 expression and the transcriptional activity induced by HEME. As expected, silencing Nrf2 expression negated HEME’s ability to increase GSH levels and enhance the antioxidant enzyme activities in SK-N-SH-MJD78 cells. Notably, silencing Nrf2 blocked the therapeutic effects of HEME on mutant polyQ-expanded ataxin-3 expression and aggregation, as well as on Hsp27 expression and autophagy in SK-N-SH-MJD78 cells (*p* < 0.0001, Figure 8).

## 4. Discussion

Although no viable medical management currently exists to effectively treat SCA3, data from various in vivo and in vitro models demonstrate that reducing the amount and aggregation of mutant polyQ-expanded ataxin-3 may delay or slow the onset and progression of neurodegeneration in SCA3 [4,5]. Moreover, mutant polyQ-expanded proteins such as ataxin-3 can reduce Hsp27 expression, which has neuroprotective potential by preventing protein aggregation [24,25] and countering increased ROS production and decreased GSH levels [26,27]. In addition to reducing the expression and aggregation of mutant polyQ-expanded ataxin-3, the present data showed that HEME decreased ROS levels and increased Hsp27 protein expression and GSH levels in both cell and *Drosophila* models of SCA3, which express either full-length or truncated mutant ataxin-3 with 78 glutamine residues. Most importantly, HEME treatment positively affected the longevity and locomotor activity of ELAV-SCA3tr-Q78 flies, which were reduced compared to ELAV-MJDtr-Q27 control flies.

Additionally, studies have focused on other molecular mechanisms associated with the pathogenesis of polyQ diseases, which may serve as potential targets for disease-modifying treatments. Enhancing autophagy and boosting antioxidant capacity, which are compromised by mutant polyQ proteins, have revealed substantial benefits for neuronal health in polyQ diseases such as Huntington’s disease and SCA3 [1,5,6]. Previous data have shown that the mammalian target of rapamycin inhibitor temsirolimus and n-Butylidenephthalide, through the upregulation of autophagy, reduced polyQ disease-causing protein aggregates in brain neurons and alleviated motor impairment in mouse models of Huntington’s disease and SCA3 [10,11,28]. Considering the data, increases in autophagy protein expression and stained cells with a lysosomotropic agent induced by HEME may play a significant role in mitigating the neurotoxic effects of mutant polyQ-expanded ataxin-3.

Compared with controls, SCA3 patients exhibit lower levels of thiol protein and activity of antioxidant enzymes, such as SOD and GSH-PX, as well as higher oxidative stress and increased mitochondrial and nuclear DNA damage [29,30,31]. Although it remains an open question whether increasing cellular antioxidant capacity is a viable therapeutic strategy for treating SCA3 patients, in vitro and in vivo models of SCA3 and other polyQ diseases have shown that upregulating antioxidant defenses can counteract neurodegeneration [1]. In addition to reducing neuronal death caused by the pro-oxidant [21], the data presented herein demonstrated that HEME treatment increased the antioxidant defense capacity in both cell and *Drosophila* models of SCA3, leading to improved lifespan and locomotor activity in SCA3 flies. In conclusion, HEME not only decreased ROS-induced neurotoxicity but also increased antioxidant capacity, which is crucial for neuronal cell health in the experimental models of SCA3.

The Nrf2/ARE pathway is well established as a central component of the cell survival response due to its role in protecting against various toxic and oxidative insults by inducing the expression of numerous antioxidant enzymes, including HO-1, NQO-1, GPx, SOD, catalase, and GR [13]. Remarkably, Nrf2 increases the transcriptional expression of p62, which is crucial for the autophagic removal of protein aggregates and damaged intracellular organelles [32,33]. p62 can then interact with Keap1, disrupting the binding between Keap1 and Nrf2, which leads to the stabilization and translocation of Nrf2 to the nucleus, where it activates its transcriptional activity [32]. Apart from the Nrf2/ARE and Nrf2-p62 autophagy pathways, previous data showed that in cells expressing mutant polyQ-expanded ataxin-3 with 75 glutamine repeats, overexpression of Nrf2 decreases the aggregation of mutant polyQ-expanded ataxin-3, whereas Nrf2 knockdown increases it [15]. The present data showed that HEME treatment increased total Nrf2 expression in ELAV-SCA3tr-Q78 flies and notably enhanced nuclear Nrf2 expression and transcriptional activity in SK-N-SH-MJD78 cells. Disrupting Nrf2 expression with si-Nrf2 revealed that Nrf2 activation by HEME is essential for its treatment efficacy, as it not only enhances antioxidant potency but also alleviates levels of mutant polyQ-expanded ataxin-3 and protein aggregates in SK-N-SH-MJD78 cells.

A review of epidemiological studies has revealed that consuming any type of mushroom in the diet provides significant benefits for cognition and mood in both healthy and compromised populations [34]. However, while most intervention studies on various mushroom species did not show benefits for brain health, supplementation with at least 3 g/day of *Hericium erinaceus* fruiting bodies showed some improvement in mood and cognitive function in middle-aged and older adults [34,35,36]. Moreover, the safety, tolerability, and cognitive benefits of HEME supplementation have been clinically proven in patients with mild Alzheimer’s disease [37]. Preclinical data from our group and others have demonstrated several molecular mechanisms involved in the neurotrophic and neuroprotective effects of *Hericium erinaceus* in neurodegenerative diseases, including SCA3. These mechanisms include the upregulation of antioxidant activity and Nrf2 activation [18,21,38]. Based on previous and current data from our group, HEME supplementation may be a promising candidate for clinical trials aimed at improving symptoms of SCA3 and other polyQ diseases.

## Figures and Tables

**Figure 1 antioxidants-13-01495-f001:**
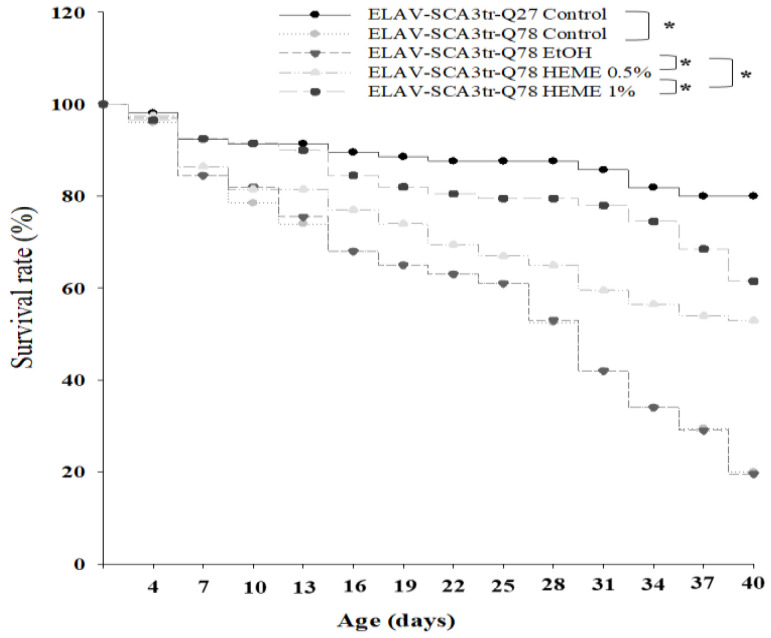
Effects of HEME on longevity in ELAV-SCA3tr-Q78 transgenic *Drosophila*. Longevity was compared across groups by Kaplan–Meier log rank analysis. The *X*-axis represents the mean life-span, and SDs are shown, * *p* < 0.001 (*n* = 320).

**Figure 2 antioxidants-13-01495-f002:**
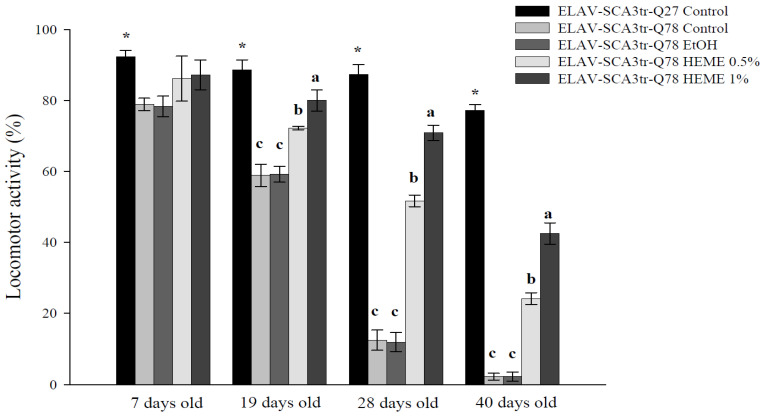
Effects of HEME on locomotor activity in ELAV-SCA3tr-Q78 transgenic Drosophila. * Values are means ± SD, *n* = 40 flies in three separate experiments. An asterisk (*) indicates a significant difference between ELAV-SCA3tr-Q27 flies and ELAV-SCA3tr-Q78 flies. In ELAV-SCA3tr-Q78 flies within the same age group, values not sharing the same letter are significantly different (*p* < 0.001).

**Figure 3 antioxidants-13-01495-f003:**
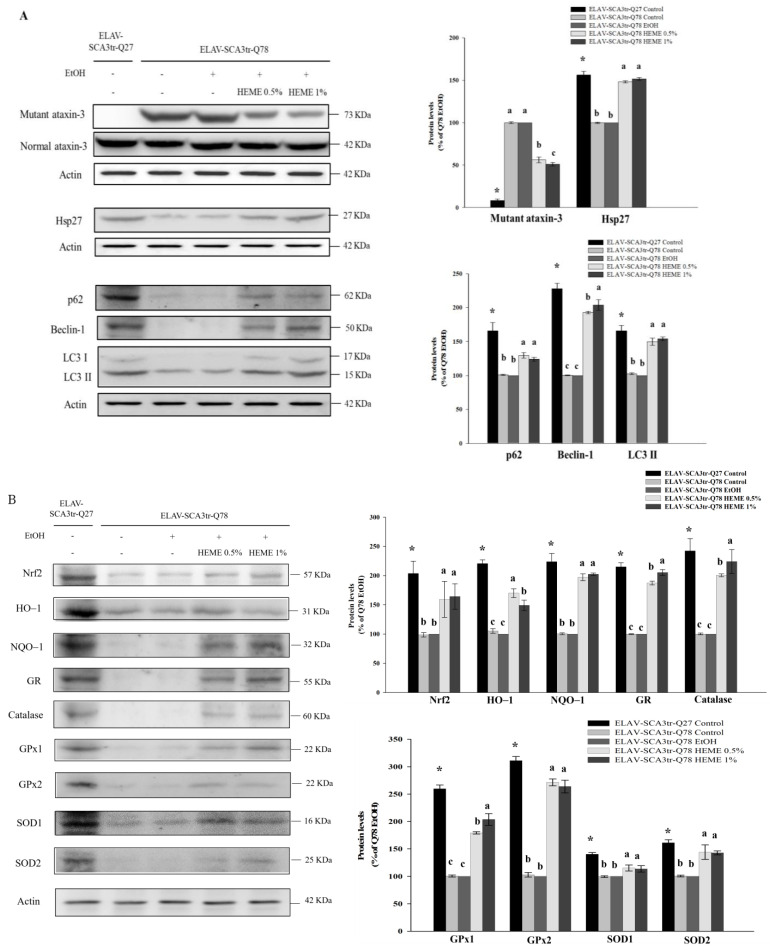
Effects of HEME on ELAV-SCA3tr-Q78 transgenic *Drosophila*. (**A**,**B**) Mutant and normal ataxin-3, Hsp27, p62, Beclin-1, LC3-II, Nrf2, HO-1, NQO1, GR, catalase, GPx1, GPx2, SOD1, and SOD2 protein expression in 19-day-old ELAV-SCA3tr-Q27 and ELAV-SCA3tr-Q78 flies. Values are means ± SD, *n* = 50 flies in three separate experiments. The values are expressed as the percentage of ELAV-SCA3tr-Q78 flies treated with the vehicle control. An asterisk (*) indicates a significant difference between ELAV-SCA3tr-Q27 flies and ELAV-SCA3tr-Q78 flies. In ELAV-SCA3tr-Q78 fly groups, values not sharing the same letter are significantly different (*p* < 0.001).

**Figure 4 antioxidants-13-01495-f004:**
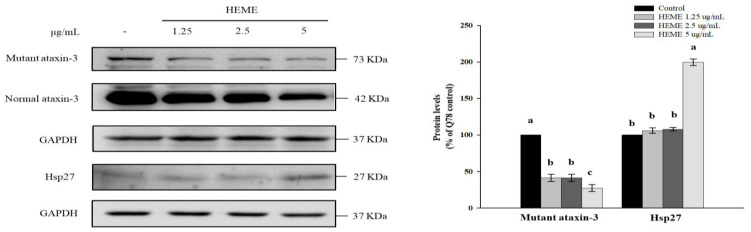
Effect of HEME on protein expression of mutant ataxin-3 and Hsp27 in SK-N-SH-MJD78 cells. Cells were treated with or without DMSO vehicle control or HEME for 24 h. Data are means ± SDs of at least three independent experiments and are expressed as the percentage of SK-N-SH-MJD78 cells treated with the vehicle control. Values not sharing the same letter are significantly different (*p* < 0.0001).

**Figure 5 antioxidants-13-01495-f005:**
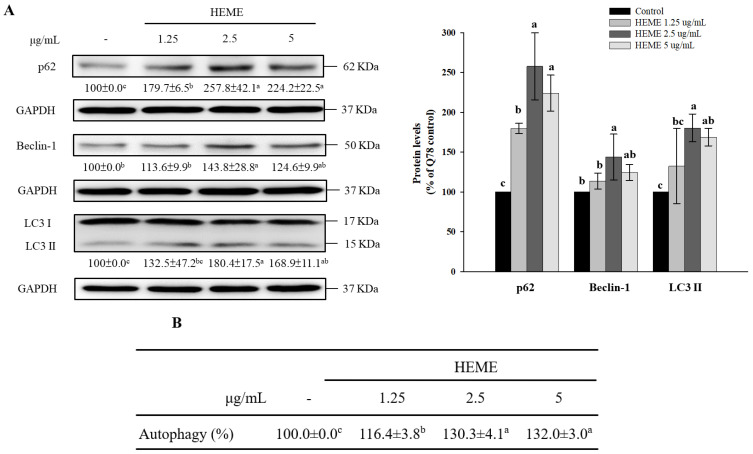
Effect of HEME on autophagy induction in SK-N-SH-MJD78 cells. Cells were treated with or without DMSO vehicle control or with 1.25–5 µg/mL HEME for 24 h. (**A**) Protein expression of p62, Beclin-1, and LC3-II. (**B**) Autophagic cells were assessed using acridine orange staining. Data are means ± SDs of at least three independent experiments and are expressed as the percentage of SK-N-SH-MJD78 cells treated with the vehicle control. Values not sharing the same letter are significantly different (*p* ≤ 0.0001).

**Figure 6 antioxidants-13-01495-f006:**
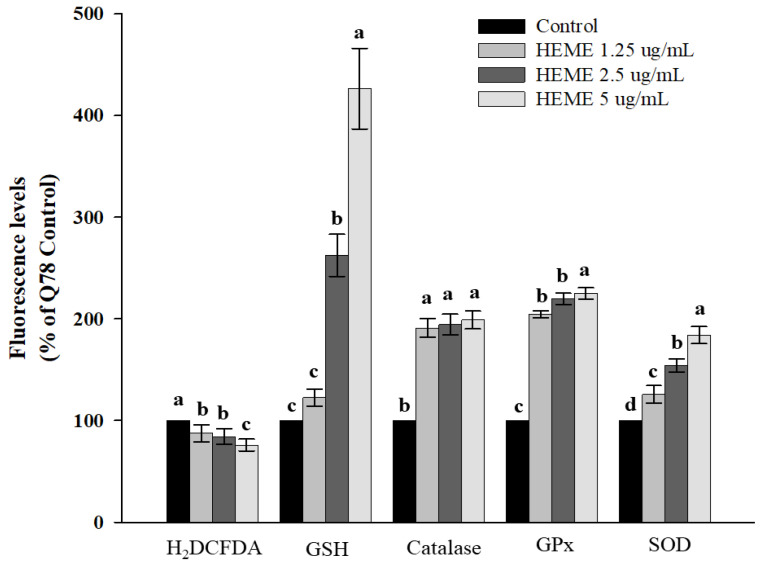
Effects of HEME on ROS, GSH levels, catalase, GPx and SOD activity in SK-N-SH-MJD78 cells. SK-N-SH-MJD78 cells were treated with or without DMSO vehicle control or HEME for 24 h, except for measurement of H2DCFDA (30 h treatment). Values are expressed as the percentage of SK-N-SH-MJD78 cells treated with the vehicle control. Data are means ± SD of at least three separate experiments and not sharing the same letter are significantly different (*p* < 0.01 and H2DCFDA; *p* = 0.012).

**Figure 7 antioxidants-13-01495-f007:**
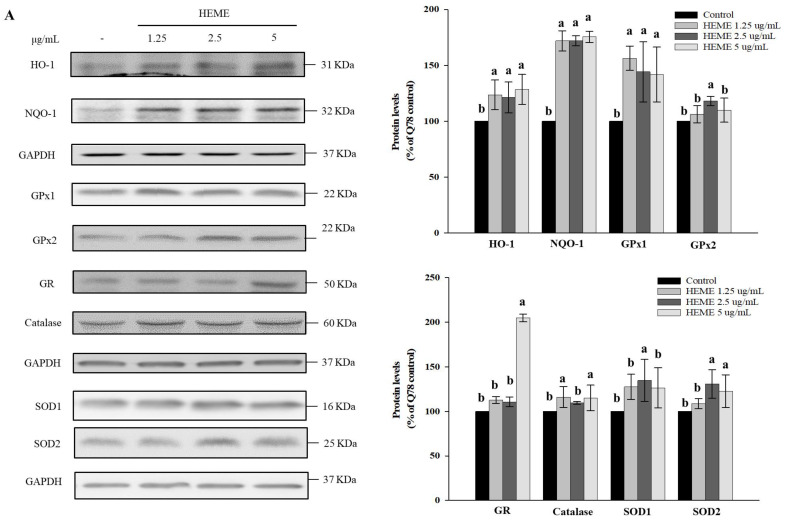
Effect of HEME on antioxidant enzyme expression and Nrf2 activation in SK-N-SH-MJD78 cells. (**A**,**B**) Cells were treated with or without DMSO vehicle control or with HEME for 24 h. Protein expression of HO-1, NQO1, GPx1, GPx2, GR, catalase, SOD1, SOD2, and nuclear Nrf2. (**C**) Before treatment with HEME, cells were transfected with ARE-luciferase reporter construct for 16 h. Data are means ± SD of at least three independent experiments and are expressed as the percentage of SK-N-SH-MJD78 cells treated with the vehicle control. Values not sharing the same letter are significantly different (*p* < 0.0001).

**Figure 8 antioxidants-13-01495-f008:**
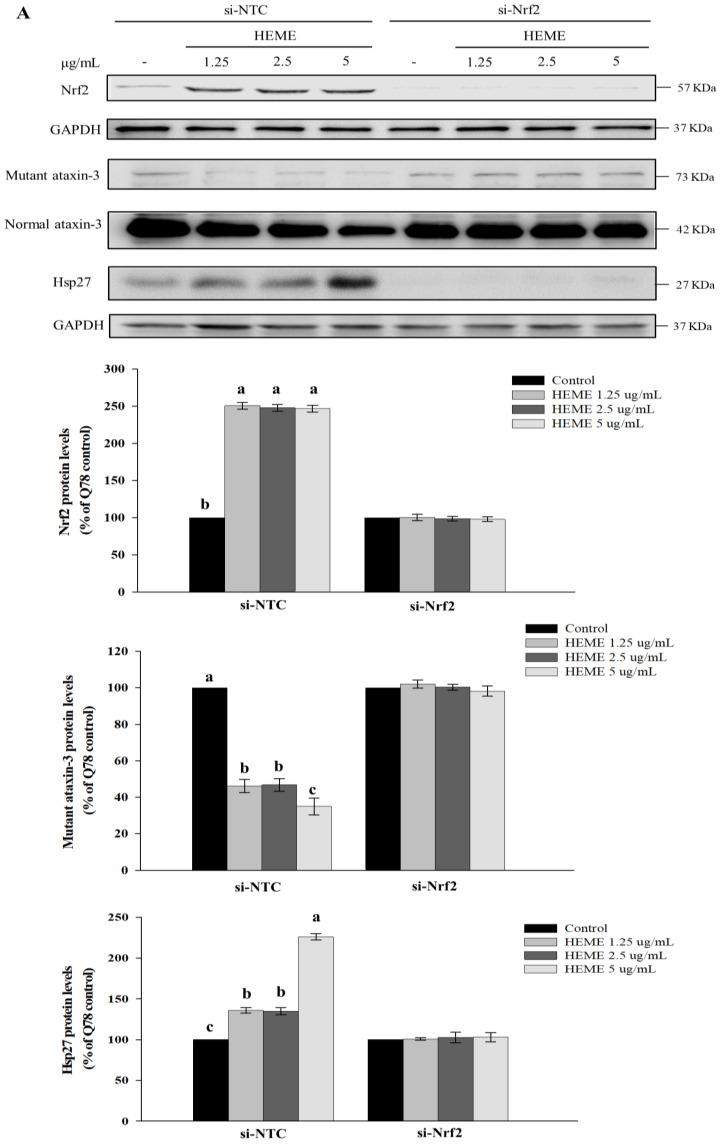
Effect of HEME in SK-N-SH-MJD78 cells transiently transfected with si-Nrf2. Cells were transiently transfected with si-NTC or si-Nrf2 as well as with or without ARE-luciferase reporter construct for 16 h and then treated with or without DMSO vehicle control or 1.25–5 µg/mL HEME for 24 h. Protein expression of (**A**) Nrf2, mutant and normal ataxin-3, and Hsp27, (**B**) p62, Beclin-1, and LC3-II. (**C**) Levels of ARE–luciferase reporter gene activities, GSH, and protein aggregations, as well as activity of catalase, GPx, and SOD. Data are means ± SD of at least three separate experiments. Within treatments with the same plasmid transfection, values are expressed as the percentage of SK-N-SH-MJD78 cells treated with the vehicle control, and values not having the same letter are significantly different (*p* < 0.0001).

**Table 1 antioxidants-13-01495-t001:** Effects of HEME on protein aggregation and ROS levels in ELAV-SCA3tr-Q78 transgenic *Drosophila* *.

	ELAV-SCA3tr-Q27	ELAV-SCA3tr-Q78			
	Control	Control	EtOH	HEME 0.5%	HEME 1%
Protein aggregation ^#^	2.3 ± 0.1 *	101.2 ± 2.0 ^a^	100.0 ± 0.0 ^a^	25.8 ± 3.5 ^b^	18.3 ± 4.1 ^c^
H_2_DCFDA ^#^	1.1 ± 0.2 *	100.4 ± 3.1 ^a^	100.0 ± 0.0 ^a^	45.8 ± 2.6 ^b^	35.0 ± 2.5 ^c^
MitoSOX ^#^	1.1 ± 0.1 *	97.6 ± 1.2 ^a^	100.0 ± 0.0 ^a^	25.7 ± 1.1 ^b^	20.5 ± 1.2 ^c^

* Levels of protein aggregation, H_2_DCFDA, and MitoSOX were measured in 19-day-old flies. ^#^ Values are means ± SD, *n* = 30 flies in three separate experiments. Values are expressed as the percentage of ELAV-SCA3tr-Q78 flies treated with the vehicle control. Values not sharing the same letter are significantly different (*p* < 0.0001).

**Table 2 antioxidants-13-01495-t002:** Effects of HEME on cell viability (MTT assay) and protein aggregation in SK-N-SH-MJD78 cells *.

			HEME	
μg/mL	-	1.25	2.5	5
MTT ^#^	100.0 ± 0.0	99.2 ± 2.1	97.0 ± 3.8	96.8 ± 1.3
Protein aggregation ^#^	100.0 ± 0.0 ^a^	43.3 ± 5.8 ^b^	16.9 ± 1.3 ^c^	12.4 ± 3.2 ^c^

* SK-N-SH-MJD78 cells were treated with or without DMSO vehicle control or HEME for 24 h. ^#^ Values are expressed as the percentage of SK-N-SH-MJD78 cells treated with the vehicle control. Data are means ± SDs of at least three separate experiments, and values not sharing the same letter are significantly different (*p* < 0.0001).

## Data Availability

The data presented in this study are available upon request from the corresponding author.

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
