# Peer review of "Erinacine A-Enriched *Hericium erinaceus* Mycelium Ethanol Extract Lessens Cellular Damage in Cell and *Drosophila* Models of Spinocerebellar Ataxia Type 3 by Improvement of Nrf2 Activation"

_antioxidants, 2024, doi:10.3390/antiox13121495_

Round 1
Reviewer 1 Report
This research addressed the main question whether Hericium erinaceus mycelium ethanol extract may diminish the effect of neurodegenrative diseases. I do consider it original and relevant. There are no researches estimating the effect of the fingi on the the diminsihing of the neurodegenerative diseases. The conclusions in manuscript are
consistent with the evidence and arguments presented and address the main question posed. Simply said, the study points to a novel direction on diminishing neurodegenaritive diseases. All the references appropriate and I don't have additional comments on the tables and figures. I have no remarks about study limitations, the study says it all.
Regarding to English language and style, I think Minor editing of English language required.
This research addressed the main question whether Hericium erinaceus mycelium ethanol extract may diminish the effect of neurodegenrative diseases. I do consider it original and relevant. There are no researches estimating the effect of the fingi on the the diminsihing of the neurodegenerative diseases. The conclusions in manuscript are
consistent with the evidence and arguments presented and address the main question posed. Simply said, the study points to a novel direction on diminishing neurodegenaritive diseases. All the references appropriate and I don't have additional comments on the tables and figures. I have no remarks about study limitations, the study says it all.
Regarding to English language and style, I think Minor editing of English language required.
Author Response
Please refer to the attached file for all responses to the reviewer's comments

Reviewer 2 Report
In the article, the data a showed that treatment with erinacine A–enriched Hericium erinaceus mycelium ethanol extract (HEME) extended longevity and improved locomotor activity in ELAV-SCA3tr-Q78 transgenic Drosophila. HEME treatment enhanced antioxidant potency and autophagy, which in turn corrected levels of mutant polyQ-expanded ataxin-3 and restrained protein aggregation in both cell and Drosophila models. Silencing Nrf2 protein expression negated most of the promising effects of HEME on SK-N-SH-MJD78 cells and showed the critical role of increased Nrf2 activation in the efficacy of HEME treatment. HEME has therapeutic potential in SCA3 by enhancing autophagic and Nrf2-mediated antioxidant pathways and may influence neurodegenerative progression in other polyQ diseases.
Regarding the article, I have next comments and recommendations:
· In Materials and Methods, I recommend adding USA at the end of the data on the origin of the chemicals used, like L.137: (Beverly, MA, USA), L. 141: (Cambridge, MA, USA), L. 155-156: (Systat Software, Inc., San Jose, CA, USA), L. 164-165: (HyClone Laboratories, Logan, UT, USA), L. 166: (Thermo Fisher Scientific, Rockford, IL, USA), L. 183-184: (New Life Science Product, Inc., Boston, MA, USA), L. 185: (Bio-Rad, Hercules, CA, USA), L. 186-187: (Perkin–Elmer Life Science, Boston, MA, USA), L. 187-188: (LAS-1000 187 plus, Fuji Photo Film Company, Minato, Japan), L. 187-188: (Alpha Innotech Corp., San Leandro, CA, USA), L. 196: (Applied Biosystems, Foster City, CA, USA), L. 209: (BMG LabTech, Ortenberg, Germany), L. 221: Promega Co. (Wisconsin, MA, USA).
· Between number and °C should be a space like 25 °C.
· L. 151: 2.3. Longevity and locomotor activity; L. 235: 3.1. Effects of HEME on longevity and locomotor activity in ELAV-SCA3tr-Q78 transgenic Drosophila.
· Figure 1: description of vertical axis should be: Survival rate (%); in Table 1: Locomotor activity (%); in Table 2: Protein aggregation; in Figure 2A the first column: Mutant ataxin-3, Mutant ataxin-3, Actin (3x), in Figure 2B: Catalase, Actin, in Figure 3: Mutant ataxin-3, Normal ataxin-3, in Table 3: Protein aggregation, in Figure 4: Autophagy, in Table 4: Catalase, in Figure 5A: Catalase, in Figure 5B: Nuclear Nrf2, in Figure 6A: Mutant ataxin-3, Normal ataxin-3, in Figure 5C: Catalase, Protein aggregation.
· L. 293: …levels and enhanced…
· L. 451: 3 g/day of Hericium erinaceus …
· In References: L. 538: J Agric Food Chem.
Author Response

(The authors gave the same response as above.)

Reviewer 3 Report
The author presented a compelling study of the effect of a natural product extract on activating the Nrf2 antioxidant transcription factor and inducing autophagy, both of which increase the survival rate of cellular and animal models. However, several inconsistencies in the manuscript prevent it from being accepted for publication in its current form.
The introduction section of this manuscript is highly similar to a previous publication (Reference 21 (529-531) of the same group of authors, specifically the first and last paragraph. Although it is understood that they continue working on the same topic using the same tools, additional effort is needed in the current manuscript to avoid suggestions for self-plagiarism.
Given that most of the results are based on Western blot measurements of the specific proteins, having the specific catalog number reference of each antibody used and mentioned in the Materials and Methods section Lines 131 to 141, 185 would be ideal.
Overall, the Materials and Methods section lacks detailed descriptions of the quantities of most of the experiments, such as the number of cells and flies or the concentrations of proteins and plasmids used in each of the experiments. This information is required for the rigor and reproducibility of this research to enhance its validity.
Several of the results data presented in tables can be better visualized by using bar graphs. The statistic description can confuse the reader, and it poorly describes the outcomes of the results. Stratification of the p-value of the statistical analysis will inform of higher significant differences with very low p-values in the results rather than simplifying as statistically significant.
There is a major inconsistency in the Western blot of the effect of HEME in SK-N-SH-MJD78 in Figure 6. The protein profile concerning the band thickness of mutant and normal ataxin-3 is entirely different in the opposite way to the one previously reported in reference 21 (529-531) by the same group of authors on the same cells. Moreover, there is no explanation for why the mutant ataxin-3 has a molecular weight of 73 kDa in any section of the manuscript.
There is also a major concern about the Western blots of Nrf2 in both Figures 2B, 5, and 6. The Western blots of the nuclear Nrf2 in Figure 5 have a high background with multiple bands above and below the marked band of 57 kDa (original western blot images1130925 Figure 5B). The author claims an increase in the Nrf2 band in response to the dose-dependent HEME treatment. However, all the background bads decrease in the negative control (left line of the Western blot), suggesting a smaller amount of protein loaded. The opposite is observed in Figure 2b not only for the Nrf2 western blot but for all the other proteins tested in which the ELAV-SCA3tr-Q27 normal has a higher background than the ELAV-SCA3tr-Q78 transgenic Drosophila. For Figure 6, an incomplete Western blot was provided that does not allow the evaluation of the antibody background.
Author Response

(The authors gave the same response as above.)

Reviewer 4 Report
The present manuscript was well organized and acceptable after minor revisions.
There were some spelling errors in the present manuscript.
The results of western blot would be better to be shown as quantitative bar graphs.
Line 285, "antioxidant enzyme" would be better to give specific names.
Line 288, correct "Figure 3A" to "Figure 2A".
Results 3.4., Figure 3 and Table 3; how about the results of SK-N-SH-WT or SK-N-SH-MJD26 cells? In other factors, the results of SK-N-SH-MJD78 compared to SK-N-SH-WT and SK-N-SH-MJD26 were written in the text. In addition, was cell viability changed in SK-N-SH-MJD78 cells compared to SK-N-SH-WT because of the increased protein aggregation, associated with the data in Table 3?
Line 359, after "........and Nrf2 nuclear protein"; the numbers of Figure should be added.
Author Response

(The authors gave the same response as above.)

Round 2
Reviewer 2 Report
In the article, the data a showed that treatment with erinacine A–enriched Hericium erinaceus mycelium ethanol extract (HEME) extended longevity and improved locomotor activity in ELAV-SCA3tr-Q78 transgenic Drosophila. HEME treatment enhanced antioxidant potency and autophagy, which in turn corrected levels of mutant polyQ-expanded ataxin-3 and restrained protein aggregation in both cell and Drosophila models. Silencing Nrf2 protein expression negated most of the promising effects of HEME on SK-N-SH-MJD78 cells and showed the critical role of increased Nrf2 activation in the efficacy of HEME treatment. HEME has therapeutic potential in SCA3 by enhancing autophagic and Nrf2-mediated antioxidant pathways and may influence neurodegenerative progression in other polyQ diseases.
Should be: 2.3. Longevity and locomotor activity; in ELAV-SCA3tr-Q27 and ELAV-SCA3tr-Q28 transgenic Drosophila
In the text and References the Latin name should be: Hericium erinaceus - see in Italics.
Author Response
Please refer to the attached file for all responses to the reviewer's comments.

Reviewer 4 Report
The authors addressed the reviewer's comments.
Some spelling errors were remaining.
Author Response

(The authors gave the same response as above.)

Round 3
Reviewer 4 Report
The authors addressed the reviewer's comments.
Spelling check could be performed at the Proof step.